# Performance Analysis of Mixed Rayleigh and $\mathcal{F}$ Distribution RF-FSO Cooperative Systems with AF Relaying

**Liqiang Han \*** , **Xuemei Liu** , **Yawei Wang and Boyu Li**

School of Electrical Engineering, Yanshan University, Qinhuangdao 066004, China;
xmliu@stumail.ysu.edu.cn (X.L.); yawei@stumail.ysu.edu.cn (Y.W.); liboyu@stumail.ysu.edu.cn (B.L.)
\* Correspondence: hanliqiang@ysu.edu.cn

**Abstract:** This paper proposes a dual-hop mixed radio frequency-free-space optical (RF-FSO) cooperative system with amplify-and-forward (AF) relaying. The RF link is subject to Rayleigh fading and the FSO link is assumed to follow $\mathcal{F}$-distributed fading with pointing error. The different types of detection, including intensity modulation with direct detection (IM/DD) and coherent heterodyne detection (HD), are considered. The closed-form expressions for the cumulative distribution function (CDF), the moment generating function (MGF), and the moments of end-to-end signal-to-noise ratio (SNR) are derived. Furthermore, closed-form expressions for the outage probability, average bit error rate (BER), and ergodic capacity are presented. Asymptotic outage probability expressions are derived to obtain additional insights into the system performance. It is shown that the HD technique exhibits better performance than an IM/DD technique. The system performance is deteriorated by atmospheric turbulence and pointing error. Finally, analytical results are confirmed by Monte Carlo simulations.

**Keywords:** $\mathcal{F}$-distributed fading; mixed RF-FSO; AF relaying; outage probability; BER; ergodic capacity

## 1. Introduction

Due to higher data rates and bandwidth, lower power consumption, free spectrum licensing, and ease of deployment, the FSO communication system has gained increasing attention [1–4]. However, some drawbacks still exist in the FSO communication system, such as the requirement of line-of-sight (LOS) and performance loss induced by atmospheric turbulence, sensitivity to obstruction, attenuation, and misalignment [5]. Compared with the FSO communication system, RF is not sensitive to atmospheric turbulence and supplies higher mobility and better communication performance in non-line-of-sight (NLOS) conditions [6]. To merge the best features of RF and FSO link, mixed RF-FSO cooperative systems are provided in various works to reduce the impact of atmospheric turbulence and the requirement for LOS, and improve the system performance significantly [7]. As an intermediate, the cooperative relay is deployed in the RF-FSO system to transmit the signal received from the transmitter to the receiver. The relay strategies can be classified as amplify-and-forward (AF) relaying and decode-and-forward (DF) relaying protocols. AF relaying is widely used due to its simplification as compared with DF relaying, in which AF relaying simply amplifies the optical signal while DF decodes the source signal before the signal is forwarded to the optical receiver. Thus, the processing overhead is significantly higher in the case of the DF relaying technique, which increases the overall cost of the communication system.

For mixed RF-FSO relaying systems, there are many fading models used in the RF channel and FSO channel. The RF link can be modeled by Rayleigh [8,9], Nakagami-m [10–12], generalized $K - \mu$ [13], and K distribution [14], and the FSO link can be characterized by Gamma-Gamma [8,9,12,15], $\mathcal{M}$ [16], exponentiated Weibull [10,13], and so on. In [8], the AF relay is used for a mixed RF/FSO relaying system, in which the RF and FSO link are described by

Rayleigh distribution and Gamma-Gamma distribution, respectively. In [10], it is supposed that the RF link is modeled by Nakagami-m distribution and the FSO link follows exponentiated Weibull fading, in which a fixed-gain relay scheme and variable-gain relay scheme are utilized. A dual-hop mixed RF/FSO system is provided in [11], and the RF and FSO links follow Nakagami-m and double generalized Gamma distribution, respectively. An asymmetric dual-hop AF relaying system is considered in [12], and the RF and FSO links are modeled by Nakagami-m fading and Gamma-Gamma fading, respectively. The performance of a mixed RF-FSO system based on DF relay is analyzed in [13], in which RF hop follows $K - \mu$ shadowed fading and FSO hop adopts the exponentiated Weibull distribution. In [14], a mixed RF/FSO two-way relay (TWR) communication system is investigated, where the fading on the RF and FSO links is modeled using $\mathcal{K}$-distribution and the double Gamma turbulence model, respectively. The physical-layer security for a mixed RF/FSO system is investigated in [16], and the $\alpha - \mu$ distribution is used in the RF link and the FSO link experiences $\mathcal{M}$ fading with pointing error. Considering multiple variable-gain relays, the outage probability of a mixed RF/FSO system has been studied, in which the channel is modeled by Rayleigh fading and $\mathcal{M}$ fading with pointing errors for RF and FSO links, respectively [17]. Zhao et al. present the performance analysis of mixed RF/FSO systems with an AF relaying scheme in a Nakagami-m and exponentiated Weibull distributed channel [18]. A comprehensive performance analysis is presented in [19], where the RF link is modeled by $\eta - \mu$ fading with multiuser diversity and the FSO link follows $\mathcal{M}$-distribution fading. Yang et al. research a cooperative system with variable-gain relay, in which multiple antennas and multiple apertures are considered in source and destination [20].

Recently, there has been increasing attention given to $\mathcal{F}$ distribution [21]. For instance, $\mathcal{F}$-distributed fading was used to model a dual-hop relaying communication system, in which ideal or non-ideal hardware is considered [22]. Shankar et al. provide the performance analysis of the $\mathcal{F}$ fading model with the maximum ratio combining (MRC) scheme for device-to-device communication systems [23]. Yoo et al. utilize the $\mathcal{F}$ distribution to model the effect of multipath and shadowing in RF links [24]. The results in [24] show that the $\mathcal{F}$ fading model provides a better fit compared to the generalized-$K$ model. In contrast to generalized-$K$ and log-normal distribution, the expressions for PDFs and CDFs are less complex using the $\mathcal{F}$ distribution. More recently, $\mathcal{F}$ distribution was used in the FSO channel to model turbulence fading for further exploration [25]. From this work, we know that the $\mathcal{F}$ distribution provides an improved fit to simulation data in different practical propagation scenarios and tractable expressions compared with well-known Gamma-Gamma and $\mathcal{M}$ distributions. Moreover, in the FSO communication scenario, the authors analyze hybrid mmWave/FSO systems over the $\mathcal{F}$-distribution fading channel [26]. In order to better characterize the situation of FSO communication, the pointing error impairment is considered in the $\mathcal{F}$ distribution model [27].

Motivated by the aspects mentioned above, we propose an AF-based mixed RF-FSO communication system, where the RF channel experiences Rayleigh fading and the FSO channel is assumed to operate over $\mathcal{F}$ fading in the presence of pointing errors. The main contributions of this paper are given as follows: (i) the closed-form expressions of CDF, MGF, and the moments for AF relaying are derived; (ii) the closed-form expressions of the end-to-end performance metrics are derived, including outage probability, average BER, and ergodic capacity; (iii) asymptotic outage probability expressions are derived for both relay protocols.

The rest of the paper is organized as follows: Section 2 details the system and channel model. End-to-end SNR statistics of AF relaying are presented in Section 3. In Sections 4 and 5, the closed-form expressions for the outage probability, average BER, and ergodic capacity are derived. Asymptotic outage probability analysis is given in Section 6. Section 7 presents the simulation and analytical results. Finally, Section 8 presents the conclusions.

## 2. System and Channel Models

A dual-hop mixed RF-FSO communication system is depicted in Figure 1, which has an RF transmitter (S), an AF relay (R), and an FSO receiver (D). The S and D are equipped

with a single antenna or aperture. The relay R converts the radio waves into optical waves with the AF relaying protocol.

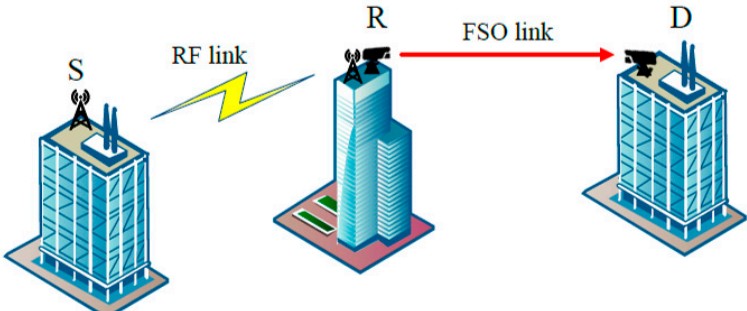

**Figure 1.** Block diagram of the RF-FSO system.

### 2.1. RF Link

The first hop is the RF link, and the received signal at R can be expressed as:

$$y_R = h_{RF}x + n_{SR} \tag{1}$$

where $h_{RF}$ is the channel coefficient, $x$ denotes the signal from the transmitter, and $n_{SR}$ is the additive white Gaussian noise (AWGN) with zero mean and $N_{01}$ variance.

The instantaneous SNR for the first hop is $\gamma_{RF} = \frac{E_s|h_{RF}|^2}{N_{01}} = |h_{RF}|^2\bar{\gamma}_{RF}$. $E_s$ is the average power and $\bar{\gamma}_{RF} = E_s/N_{01}$ denotes the average SNR. The RF link is characterized by Rayleigh fading, and the PDF and CDF of $\gamma_{RF}$ can be given as [6]:

$$f_{\gamma_{RF}}(\gamma) = \frac{1}{\bar{\gamma}_{RF}}\exp\left(-\frac{\gamma}{\bar{\gamma}_{RF}}\right) \tag{2}$$

and

$$F_{\gamma_{RF}}(\gamma) = 1 - \exp\left(-\frac{\gamma}{\bar{\gamma}_{RF}}\right) \tag{3}$$

### 2.2. FSO Link

The relay R converts the RF signal into the optical signal utilizing subcarrier intensity modulation (SIM) and forwards it to D. A direct current (DC) can be added to the signal to avoid the signal having negative values [18]. The optical signal at R can be written as:

$$S_{opt} = G\left(1 + \eta_e^{\frac{t}{2}}y_R\right) \tag{4}$$

where $\eta_e$ represents the photoelectric conversion ratio, and $G$ is the amplification gain. The value of $t$ depends on the type of detection method (i.e., $t = 1$ corresponds to HD and $t = 2$ corresponds to IM/DD). Through the FSO link, the signal at D can be written as:

$$y_D = I^{\frac{t}{2}}G\left[1 + (\eta_e)^{\frac{t}{2}}(h_{RF}x + n_{SR})\right] + n_{RD} \tag{5}$$

where $I$ denotes the channel coefficient of the FSO link. $n_{RD}$ represents AWGN with the zero mean and $N_{02}$ variance.

Filtering out the DC bias, the signal at D can be derived as:

$$y_D = (\eta_e I)^{\frac{t}{2}}G(h_{RF}x + n_{SR}) + n_{RD} \tag{6}$$

It is assumed that the FSO link is impaired by $\mathcal{F}$-distributed atmospheric turbulence $(I_a)$, pointing errors $(I_p)$, and path loss $(I_l)$, in which the channel coefficient $I = I_a I_p I_l$.

The path loss $I_l$ is given by the Beer–Lambert Law as:

$$I_l = \exp(-\sigma L) \tag{7}$$

where $\sigma$ is the attenuation coefficient and $L$ is the propagation distance.

$\mathcal{F}$ distribution consists of Gamma distribution and inverse Gamma distribution. The PDF of $I_a$ is written as [25]:

$$f_{I_a}(x) = \frac{a^a (b-1)^b x^{a-1}}{\mathcal{B}(a,b)(ax+b-1)^{a+b}} \tag{8}$$

where $\mathcal{B}(\cdot,\cdot)$ is the beta function. The parameters $a$ and $b$ describe the atmospheric turbulence conditions. Parameters $a$ and $b$ are computed as [25]:

$$a = \frac{1}{\exp(\sigma_{\ln S}^2) - 1} \quad \text{and} \quad b = \frac{1}{\exp(\sigma_{\ln L}^2) - 1} + 2 \tag{9}$$

where $\sigma_{\ln S}^2$ is the small-scale log-irradiance variance and $\sigma_{\ln L}^2$ is the large-scale log-irradiance variance. The infinite plane waves, spherical waves, and Gaussian-beam waves can be used in the corresponding analysis. Supposing spherical wave propagation, the $\sigma_{\ln S}^2$ is given by [28]:

$$\sigma_{\ln S}^2 = \frac{0.51\delta_{SP}^2 \left(1 + 0.69\delta_{SP}^{12/5}\right)^{-5/6}}{1 + 0.90d^2(\sigma_1/\delta_{SP})^{12/5} + 0.62d^2\sigma_1^{12/5}} \tag{10}$$

where $\sigma_{SP}^2$ is the spherical wave scintillation index with weak irradiance fluctuations, which is obtained by [29]:

$$\begin{aligned}
\sigma_{SP}^2 = 9.65\sigma_1^2 \Bigg\{ &0.4\left(1 + 9/Q_l^2\right)^{11/12}\left[\sin\left(\frac{11}{6}\arctan\frac{Q_l}{3}\right)\right.\\
&+ \frac{2.61}{(9+Q_l^2)^{1/4}}\sin\left(\frac{4}{3}\arctan\frac{Q_l}{3}\right) - \frac{0.52}{(9+Q_l^2)^{7/24}}\\
&\times \left. \sin\left(\frac{5}{4}\arctan\frac{Q_l}{3}\right) - \frac{3.5}{Q_l^{5/6}}\right]\Bigg\}
\end{aligned} \tag{11}$$

where $\sigma_1^2 = 0.5C_n^2\beta^{7/6}L^{11/6}$ is the Rytov variance. $C_n^2$ is the index of the refraction structure parameter. $Q_l = 10.89L/(\beta l_0^2)$. $l_0$ (mm) indicates the inner scale. $d = \sqrt{\beta D^2/4L}$ is the equivalent aperture diameter. $\beta = 2\pi/\lambda$ is the optical wave number and $\lambda$ is the wavelength. $\sigma_{\ln L}^2$ can be derived by $\sigma_{\ln L}^2 = \sigma_{\ln L}^2(l_0) - \sigma_{\ln L}^2(L_0)$ [28]. $l_0$ and $L_0$ denote the inner scale and outer scale. $\sigma_{\ln L}^2(\cdot)$ is the large-scale log-irradiance variance and the unified expression can be given by:

$$\begin{aligned}
\sigma_{\ln L}^2(u) = 0.04\sigma_1^2 &\left(\frac{\eta(u)Q_l}{\eta(u) + Q_l}\right)^{7/6} \times \\
&\left[1 + 1.75\left(\frac{\eta(u)}{\eta(u)+Q_l}\right)^{1/2} - \left(\frac{\eta(u)}{\eta(u)+Q_l}\right)^{7/12}\right]
\end{aligned} \tag{12}$$

where $u \in \{l_0, L_0\}$, $\eta(l_0) = \frac{8.56}{1+0.18d^2+0.20\sigma_1^2 Q_l^{1/6}}$, $\eta(L_0) = \frac{Q_0\eta(l_0)}{Q_0+\eta(l_0)}$, and $Q_0 = \frac{64\pi^2 L}{\beta L_0^2}$.

The PDF of pointing error impairments $I_p$ is given as [30]:

$$f_{I_p}(y) = \frac{m^2}{A_0^{m^2}}y^{m^2}, \quad 0 \leq y \leq A_0 \tag{13}$$

where $A_0 = [\text{erf}(v)]^2$ is the fraction of collected power, $\text{erf}(\cdot)$ is the error function and $v = (\sqrt{\pi}a)/(\sqrt{2}w_z)$. $w_z$ is the beam waist, and $a$ is the receiver's aperture radius. $m = w_{zeq}/2\sigma_s$ is the ratio between the beam radius and the pointing error displacement standard deviation at the receiver. $w_{zeq}^2 = w_z^2 \frac{\sqrt{\pi}\text{erf}(v)}{2v\exp(-v^2)}$. Note that $m \to \infty$ is considered as no pointing errors.

Considering the effect of atmospheric turbulence, pointing error, and path loss, the combined PDF and CDF of channel coefficient $I$ can be derived, respectively, as [27]:

$$f_I(h) = \frac{am^2 G_{2,2}^{2,1}\left[\frac{a}{(b-1)h_l A_0} h \Big|_{a-1,\, m^2-1}^{-b,\, m^2}\right]}{(b-1)h_l A_0 \Gamma(a)\Gamma(b)} \tag{14}$$

and

$$F_I(h) = \frac{m^2}{\Gamma(a)\Gamma(b)} G_{3,3}^{2,2}\left[\frac{a}{(b-1)h_l A_0} h \Big|_{J_2}^{J_1}\right] \tag{15}$$

where $J_1 = [1-b, 1, 1+m^2]$ and $J_2 = [a, m^2, 0]$. $\Gamma(\cdot)$ is the Gamma function, and $G_{p,q}^{m,n}[\cdot]$ is the Meijer's G function [31].

The instantaneous SNR for the FSO link is $\gamma_{FSO} = \frac{(\eta_e I)^t}{N_{02}}$. Applying the transformation, the PDF and CDF of $\gamma_{FSO}$ can be obtained as [27]:

$$f_{\gamma_{FSO}}(\gamma) = \frac{m^2 \gamma^{-1} G_{2,2}^{2,1}\left[\frac{\varphi_m \gamma^{1/t}}{\bar{\gamma}_{FSO}^{1/t}} \Big|_{a,\, m^2}^{1-b,\, 1+m^2}\right]}{t\Gamma(a)\Gamma(b)} \tag{16}$$

and

$$F_{\gamma_{FSO}}(\gamma) = \frac{m^2}{\Gamma(a)\Gamma(b)} G_{3,3}^{2,2}\left[\frac{\varphi_m \gamma^{1/t}}{\bar{\gamma}_{FSO}^{1/t}} \Big|_{J_2}^{J_1}\right] \tag{17}$$

where $\varphi_m = \frac{am^2}{(b-1)(1+m^2)} \cdot \bar{\gamma}_{FSO} = \frac{(\eta_e E[I])^t}{N_{02}}$ is the average SNR. $E[\cdot]$ is the expectation operator.

## 3. End-to-End SNR Statistics

The variable-gain relaying and fixed-gain relaying are considered in our work. From (6), the equivalent SNR at D can be obtained as:

$$\gamma_{eq} = \frac{E_s G^2 h_{RF}^2 (\eta_e I)^t}{G^2 (\eta_e I)^t N_{01} + N_{02}} = \frac{\frac{E_s h_{RF}^2}{N_{01}} \frac{(\eta_e I)^t}{N_{02}}}{\frac{(\eta_e I)^t}{N_{02}} + \frac{1}{G^2 N_{01}}} \tag{18}$$

For the variable-gain relaying scheme, amplifying gain $G^2 = \frac{1}{E_s h_{RF}^2 + N_{01}}$ [16]. Substituting it into (18), the equivalent instantaneous SNR can be derived as:

$$\gamma_{eq}^V = \frac{\gamma_{RF}\gamma_{FSO}}{\gamma_{RF} + \gamma_{FSO} + 1} \tag{19}$$

The fixed-gain relaying is semi-blind relaying, in which the channel state information (CSI) of the RF link is not required at the relay (i.e., the relay R has a fixed gain $G$). It is assumed that $C = 1/(G^2 N_{01})$. Thus, the equivalent instantaneous SNR can be written as [32]:

$$\gamma_{eq}^F = \frac{\gamma_{RF}\gamma_{FSO}}{\gamma_{FSO} + C} \tag{20}$$

### 3.1. Variable-Gain Relaying

For variable-gain relay, the equivalent instantaneous SNR can be calculated by using the upper bound [8]:

$$\gamma_{eq}^V = \frac{\gamma_{RF}\gamma_{FSO}}{\gamma_{RF} + \gamma_{FSO} + 1} \cong \min(\gamma_{RF}, \gamma_{FSO}) \tag{21}$$

Based on (21), the CDF for variable-gain relay is expressed as:

$$\begin{aligned} F_{\gamma_{eq}^V}(\gamma) &= \Pr(\min(\gamma_{RF}, \gamma_{FSO}) < \gamma) \\ &= F_{\gamma_{RF}}(\gamma) + F_{\gamma_{FSO}}(\gamma) - F_{\gamma_{RF}}(\gamma)F_{\gamma_{FSO}}(\gamma) \end{aligned} \tag{22}$$

Substituting (3) and (17) into (22), the final expression of the CDF for variable-gain relay is derived as:

$$F_{\gamma_{eq}^V}(\gamma) = 1 - \exp\left(-\frac{\gamma}{\bar{\gamma}_{RF}}\right)\left(1 - \frac{m^2}{\Gamma(a)\Gamma(b)} \times G_{3,3}^{2,2}\left[\frac{\varphi_m\gamma^{1/t}}{\bar{\gamma}_{FSO}^{1/t}}\Big|_{J_2}^{J_1}\right]\right) \tag{23}$$

The MGF can be computed by $\mathcal{M}_\gamma(s) = E[e^{-\gamma s}]$. After integral operation, the MGF can be rewritten in terms of the CDF as:

$$\mathcal{M}_\gamma(s) = s \int_0^\infty e^{-\gamma s} F_\gamma(\gamma)d\gamma \tag{24}$$

Substituting (23) into (24), and employing [33] (Equation (11)) and [34] (Equation (07.34.21.0013.01)), the MGF for variable-gain relay can be obtained as:

$$\mathcal{M}_\gamma(s) = 1 - \frac{s}{s + 1/\bar{\gamma}_{RF}} + \frac{sm^2}{\Gamma(a)\Gamma(b)} \times \frac{t^{a+b-2}}{(2\pi)^{t-1}(s + 1/\bar{\gamma}_{RF})}$$
$$\times G_{3t+1,3t}^{2t,2t+1}\left[\frac{\varphi_m^t}{\bar{\gamma}_{FSO}(s + 1/\bar{\gamma}_{RF})}\Big|_{Q_2}^{Q_1,\Delta(1,0)}\right] \tag{25}$$

where $\Delta(\varrho,\tau) = \left[\frac{\tau}{\varrho}, \frac{\tau+1}{\varrho}, \cdots, \frac{\tau+\varrho-1}{\varrho}\right]$. $Q_1 = [\Delta(t,1-b),\Delta(t,1),\Delta(t,1+m^2)]$, $Q_2 = [\Delta(t,a),\Delta(t,m^2),\Delta(t,0)]$.

The moment is defined as $E[\gamma^n]$, which can be derived by using the complementary CDF (CCDF) as given by:

$$E[\gamma^n] = n\int_0^\infty \gamma^{n-1}F_\gamma^c(\gamma)d\gamma \tag{26}$$

Substituting (23) into (26), and employing [33] (Equation (11)), ref. [34] (Equation (07.34.21.0013.01)), and the definition of $\Gamma(x) = \int_0^\infty t^{x-1}e^{-t}dt$, the moments for variable-gain relay can be derived as:

$$E[\gamma^n] = n\bar{\gamma}_{RF}^n\Gamma(n) - \frac{nm^2}{\Gamma(a)\Gamma(b)} \times \frac{t^{a+b-2}\bar{\gamma}_{RF}^n}{(2\pi)^{t-1}}$$
$$\times G_{3t+1,3t}^{2t,2t+1}\left[\frac{\varphi_m^t\bar{\gamma}_{RF}}{\bar{\gamma}_{FSO}}\Big|_{Q_2}^{Q_1,\Delta(1,1-n)}\right] \tag{27}$$

### 3.2. Fixed-Gain Relaying

The CDF for fixed-gain relaying is expressed as:

$$F_{\gamma_{eq}^F}(\gamma) = \Pr\left[\frac{\gamma_{RF}\gamma_{FSO}}{\gamma_{FSO} + C} < \gamma\right]$$
$$= \int_0^\infty \Pr\left[\frac{\gamma_{RF}\gamma_{FSO}}{\gamma_{FSO} + C} < \gamma|\gamma_{FSO}\right]f_{\gamma_{FSO}}(\gamma_{FSO})d\gamma_{FSO} \tag{28}$$
$$= \int_0^\infty F_{\gamma_{RF}}\left[\frac{(\gamma_{FSO} + C)\gamma}{\gamma_{FSO}}\right]f_{\gamma_{FSO}}(\gamma_{FSO})d\gamma_{FSO}$$

Substituting (3) and (16) into (28), the CDF for fixed-gain relaying is derived as:

$$F_{\gamma_{eq}^F}(\gamma) = 1 - \frac{m^2}{t\Gamma(a)\Gamma(b)}\exp(-\gamma/\bar{\gamma}_{RF})\int_0^\infty \gamma_{FSO}^{-1}e^{-\frac{\gamma C}{\gamma_{FSO}\bar{\gamma}_{RF}}}$$
$$\times G_{2,2}^{2,1}\left[\frac{\varphi_m\gamma^{1/t}}{\bar{\gamma}_{FSO}^{1/t}}\Big|_{a,m^2}^{1-b,1+m^2}\right]d\gamma_{FSO} \tag{29}$$

Using [33] (Equation (11)) and [31] (Equation (9.31.2)), the exponential function in (29) can be expressed as:

$$\exp\left(-\frac{\gamma C}{\gamma_{FSO}\bar{\gamma}_{RF}}\right) = G_{1,0}^{0,1}\left[\frac{\bar{\gamma}_{RF}\gamma_{FSO}}{C\gamma}\Big|_{-}^{1}\right] \tag{30}$$

Therefore, the integral in (29) can be solved by employing the exponential function expressed above and [34] (Equation (07.34.21.0013.01)). The CDF of fixed-gain relay can be derived as:

$$F_{\gamma_{eq}^F}(\gamma) = 1 - A_1 \exp(-\gamma/\bar{\gamma}_{RF}) G_{2t,2t+1}^{2t+1,t} \left[ \frac{C\gamma\varphi_m^t}{\bar{\gamma}_{RF}\bar{\gamma}_{FSO}} \Big| \begin{matrix} B_1 \\ B_2 \end{matrix} \right] \tag{31}$$

where $A_1 = \frac{m^2}{\Gamma(a)\Gamma(b)} \times \frac{t^{a+b-2}}{(2\pi)^{t-1}}$ and $B_1 = \left[ \Delta(t, 1-b), \Delta(t, 1+m^2) \right]$, $B_2 = \left[ \Delta(t, a), \Delta(t, m^2), \Delta(1, 0) \right]$.

Substituting (31) into (24), and employing [33] (Equation (11)) and [34] (Equation (07.34.21.0011.01)), the MGF can be derived as:

$$\begin{aligned} \mathcal{M}_\gamma(s) = 1 &- \frac{sm^2}{\Gamma(a)\Gamma(b)} \times \frac{t^{a+b-2}}{(2\pi)^{t-1}(s+1/\bar{\gamma}_{RF})} \\ &\times G_{2t+1,2t+1}^{2t+1,t+1} \left[ \frac{C\varphi_m^t}{(s+1/\bar{\gamma}_{RF})\bar{\gamma}_{RF}\bar{\gamma}_{FSO}} \Big| \begin{matrix} B_1, \Delta(1,0) \\ B_2 \end{matrix} \right] \end{aligned} \tag{32}$$

Substituting (31) into (26), and employing [33] (Equation (11)) and [34] (Equation (07.34.21.0011.01)), the moments can be derived as:

$$\begin{aligned} E[\gamma^n] = &\frac{nm^2\bar{\gamma}_{RF}^n}{\Gamma(a)\Gamma(b)} \times \frac{t^{a+b-2}}{(2\pi)^{t-1}} \\ &\times G_{2t+1,2t+1}^{2t+1,t+1} \left[ \frac{C\varphi_m^t}{\bar{\gamma}_{FSO}} \Big| \begin{matrix} B_1, \Delta(1, 1-n) \\ B_2 \end{matrix} \right] \end{aligned} \tag{33}$$

## 4. Performance Analysis for Variable-Gain Relaying

### 4.1. Outage Probability

The outage probability can be defined as the probability that the instantaneous SNR falls below a threshold, $\gamma_{th}$. The outage probability expression of the considered communication system can be expressed by replacing $\gamma$ with $\gamma_{th}$ in the CDF in (23), which is given by:

$$P_{out}^V(\gamma_{th}) = F_{\gamma_{eq}^V}(\gamma_{th}) \tag{34}$$

### 4.2. Average BER

Based on various binary modulations, the average BER can be derived by using CDF as:

$$\bar{P}_e = \frac{q^p}{2\Gamma(p)} \int_0^\infty \exp(-q\gamma)\gamma^{p-1} F_\gamma(\gamma) d\gamma \tag{35}$$

where $p$ and $q$ are determined by particular types of modulation schemes [35]. The coherent binary frequency shift keying (CBFSK), coherent binary phase shift keying (CBPSK), non-coherent binary frequency shift keying (NBFSK), and differential binary phase shift keying (DBPSK) are given by $(p, q)$ = (0.5, 0.5), (0.5, 1), (1, 0.5), (1, 1), respectively.

Inserting (23) into (35), the average BER of variable-gain relay can be expressed as:

$$\begin{aligned} \bar{P}_e^V = &\frac{q^p}{2\Gamma(p)} \int_0^\infty \exp(-q\gamma)\gamma^{p-1} d\gamma \\ &- \frac{q^p}{2\Gamma(p)} \int_0^\infty \exp\left[-\left(q+\frac{1}{\bar{\gamma}_{RF}}\right)\gamma\right]\gamma^{p-1} d\gamma \\ &+ \frac{m^2 q^p}{2\Gamma(p)\Gamma(a)\Gamma(b)} \int_0^\infty \exp\left[-\left(q+\frac{1}{\bar{\gamma}_{RF}}\right)\gamma\right]\gamma^{p-1} \\ &\times G_{3,3}^{2,2} \left[ \frac{\varphi_m\gamma^{1/t}}{\bar{\gamma}_{FSO}^{1/t}} \Big| \begin{matrix} 1-b, 1, 1+m^2 \\ a, m^2, 0 \end{matrix} \right] d\gamma \end{aligned} \tag{36}$$

Utilizing [33] (Equation (11)) and [34] (Equation (07.34.21.0013.01)), and the definition of $\Gamma(x) = \int_0^\infty t^{x-1} e^{-t} dt$, the average BER expression for variable-gain relaying is obtained as:

$$\bar{P}_e^V = \frac{1}{2} - \frac{q^p}{2(q+1/\bar{\gamma}_{RF})^p} + A_1 \frac{q^p}{2\Gamma(p)(q+1/\bar{\gamma}_{RF})^p}$$
$$\times G_{3t+1,3t}^{2t,2t+1}\left[\frac{\varphi_m^t}{\bar{\gamma}_{FSO}(q+1/\bar{\gamma}_{RF})}\bigg|\begin{matrix}Q_1,\Delta(1,1-p)\\Q_2\end{matrix}\right] \tag{37}$$

### 4.3. Ergodic Capacity

The definition of ergodic capacity is $\bar{C} = E[\log_2(1+\alpha_t\gamma)]$, where $\alpha_t = 1$ for HD and $\alpha_t = e/2\pi$ for IM/DD [35]. With the help of the CCDF $F_\gamma^c(\gamma)$, the ergodic capacity can be written as:

$$\bar{C} = \frac{\alpha_t}{\ln 2}\int_0^\infty \frac{F_\gamma^c(\gamma)}{1+\alpha_t\gamma}d\gamma \tag{38}$$

Substituting (23) into (38), the ergodic capacity is expressed as:

$$\bar{C}^V = \frac{\alpha_t}{\ln 2}\int_0^\infty \frac{1}{1+\alpha_t\gamma}\exp\left(-\frac{\gamma}{\bar{\gamma}_{RF}}\right)d\gamma$$
$$-\frac{\alpha_t m^2}{\ln 2\Gamma(a)\Gamma(b)}\int_0^\infty \frac{1}{1+\alpha_t\gamma}\exp\left(-\frac{\gamma}{\bar{\gamma}_{RF}}\right)$$
$$\times G_{3,3}^{2,2}\left[\frac{\varphi_m\gamma^{1/t}}{\bar{\gamma}_{FSO}^{1/t}}\bigg|\begin{matrix}1-b,1,1+m^2\\a,m^2,0\end{matrix}\right]d\gamma \tag{39}$$

Utilizing [31] (Equation (7.813.1)) and [33] (Equations (10) and (11)), and with the aid of the integral of the extended generalized bivariate Meijer's G function (EGBMGF) [36] (Equation (20)), the ergodic capacity expression for the variable-gain relay is derived as:

$$\bar{C}^V = \frac{\alpha_t\bar{\gamma}_{RF}}{\ln 2}G_{2,1}^{1,2}\left[\alpha_t\bar{\gamma}_{RF}\bigg|\begin{matrix}0,0\\0\end{matrix}\right] - A_1\frac{\alpha_t}{\ln 2}\bar{\gamma}_{RF}$$
$$\times G_{1,0:1,1:3t,3t}^{1,0:1,1:2t,2t}\left[\begin{matrix}1\\-\end{matrix}\bigg|\begin{matrix}0\\0\end{matrix}\bigg|\begin{matrix}Q_1\\Q_2\end{matrix}\bigg|\alpha_t\bar{\gamma}_{RF},\varphi_m^t\frac{\bar{\gamma}_{RF}}{\bar{\gamma}_{FSO}}\right] \tag{40}$$

## 5. Performance Analysis for Fixed-Gain Relaying

### 5.1. Outage Probability

For fixed-gain relaying, the outage probability is expressed by replacing $\gamma$ with $\gamma_{th}$ in (31), which is given by:

$$P_{out}^F(\gamma_{th}) = F_{\gamma_{eq}^F}(\gamma_{th}) \tag{41}$$

### 5.2. Average BER

Inserting (31) into (35), the average BER of fixed-gain relaying can be expressed as:

$$\bar{P}_e^F = \frac{q^p}{2\Gamma(p)}\int_0^\infty \exp(-q\gamma)\gamma^{p-1}d\gamma$$
$$- A_1\frac{q^p}{2\Gamma(p)}\int_0^\infty \exp(-q\gamma)\gamma^{p-1}\exp(-\gamma/\bar{\gamma}_{RF})$$
$$\times G_{2t,2t+1}^{2t+1,t}\left[\frac{C\gamma\varphi_m^t}{\bar{\gamma}_{RF}\bar{\gamma}_{FSO}}\bigg|\begin{matrix}B_1\\B_2\end{matrix}\right]d\gamma \tag{42}$$

Utilizing [33] (Equation (11)) and [34] (Equation (07.34.21.0013.01)), and the definition of $\Gamma(x)$, the expression of the average BER is derived as:

$$\bar{P}_e^F = \frac{1}{2} - A_1\frac{q^p}{2\Gamma(p)(q+1/\bar{\gamma}_{RF})^p}$$
$$\times G_{2t+1,2t+1}^{2t+1,t+1}\left[\frac{C\varphi_m^t}{\bar{\gamma}_{RF}\bar{\gamma}_{FSO}(q+1/\bar{\gamma}_{RF})}\bigg|\begin{matrix}B_1,\Delta(1,1-p)\\B_2\end{matrix}\right] \tag{43}$$

### 5.3. Ergodic Capacity

Substituting (31) into (38), the ergodic capacity for fixed-gain relay can be expressed as:

$$
\begin{aligned}
\bar{C}^F = A_1 \frac{\alpha_t}{\ln 2} \int_0^\infty \frac{1}{1 + \alpha_t \gamma} \exp\left(-\frac{\gamma}{\bar{\gamma}_{RF}}\right) \\
\times G_{2t,2t+1}^{2t+1,t}\left[\frac{C\gamma\varphi_m^t}{\bar{\gamma}_{RF}\bar{\gamma}_{FSO}}\Big|\begin{matrix}B_1\\B_2\end{matrix}\right] d\gamma
\end{aligned}
\tag{44}
$$

With the aid of [33] (Equations (10) and (11)) and the integral of EGBMGF [36] (Equation (20)), the ergodic capacity for fixed-gain relay can be written as:

$$
\bar{C}^F = A_1 \frac{\alpha_t \bar{\gamma}_{RF}}{\ln 2} \times G_{1,0:1,1:2t,2t+1}^{1,0:1,1:2t+1,t}\left[\begin{matrix}1\\-\end{matrix}\Big|\begin{matrix}0\\0\end{matrix}\Big|\begin{matrix}B_1\\B_2\end{matrix}\Big|\alpha_t\bar{\gamma}_{RF},\frac{C\varphi_m^t}{\bar{\gamma}_{FSO}}\right]
\tag{45}
$$

## 6. Asymptotic Outage Probability Analysis

For further insights into the system performance, asymptotic outage probability analysis is focused on the variable-gain relaying protocol and fixed-gain relaying protocol. It is assumed that $\bar{\gamma}_{RF} \triangleq \bar{\gamma}_{FSO}$. In the high SNR regime, the asymptotic outage probability expression can be presented as $P_{out} \approx (G_c \cdot \bar{\gamma})^{-G_d}$, where the $G_d$ is the diversity order and $G_c$ is the coding gain.

For variable-gain relaying, the asymptotic expression of outage probability can be expressed as:

$$
F_{\gamma_{eq}^v}^\infty(\gamma_{th}) = F_{\gamma_{RF}}^\infty(\gamma_{th}) + F_{\gamma_{FSO}}^\infty(\gamma_{th}) - F_{\gamma_{RF}}^\infty(\gamma_{th})F_{\gamma_{FSO}}^\infty(\gamma_{th})
\tag{46}
$$

For the RF link, $\lim_{\bar{\gamma}_{RF}\to\infty} e^{-\gamma/\bar{\gamma}_{RF}} = 1$, and the asymptotic expression for variable-gain relaying can be simplified to $F_{\gamma_{eq}^v}^\infty(\gamma_{th}) = F_{\gamma_{FSO}}^\infty(\gamma_{th})$. Using [34] (Equation (07.34.06.0006.01)), the asymptotic expression for the second hop can be expressed as:

$$
\begin{aligned}
F_{\gamma_{FSO}}^\infty(\gamma_{th}) = \sum_{k=1}^2 \frac{\prod_{j=1,\neq k}^2 \Gamma(J_{2,j} - J_{2,k}) \prod_{j=1}^2 \Gamma(1 - J_{1,j} + J_{2,k})}{\prod_{j=3}^3 \Gamma(J_{1,j} - J_{2,k}) \prod_{j=3}^3 \Gamma(1 - J_{2,j} + J_{2,k})} \\
\times \left[\frac{(b-1)(1+m^2)\bar{\gamma}_{FSO}^{1/t}}{am^2\gamma_{th}^{1/t}}\right]^{-J_{2,k}}
\end{aligned}
\tag{47}
$$

The diversity order is $G_d = \min\left\{\frac{a}{t}, \frac{m^2}{t}\right\}$ and it can be seen that $G_d$ depends on the parameters $a$ and $m$.

For fixed-gain relaying, we use [34] (Equation (07.34.06.0006.01)) to expand the Meijer's G function in (31) as:

$$
\begin{aligned}
G_{2t,2t+1}^{2t+1,t}\left[\frac{C\gamma\varphi_m^t}{\bar{\gamma}_{RF}\bar{\gamma}_{FSO}}\Big|\begin{matrix}B_1\\B_2\end{matrix}\right] = \sum_{k=1}^{2t+1} \frac{\prod_{j=1,j\neq k}^{2t+1} \Gamma(B_{2,j} - B_{2,k})}{\prod_{j=t+1}^{2t} \Gamma(B_{1,j} - B_{2,k})} \\
\times \prod_{j=1}^t \Gamma(1 - B_{1,j} + B_{2,k})\left[\frac{\bar{\gamma}_{RF}\bar{\gamma}_{FSO}}{C\gamma_{th}\varphi_m^t}\right]^{-B_{2,k}}
\end{aligned}
\tag{48}
$$

The asymptotic outage probability expression for fixed-gain relaying can be derived by inserting (48) and $\lim_{\bar{\gamma}_{RF}\to\infty} e^{-\gamma/\bar{\gamma}_{RF}} = 1$ into (31), and $G_d = \min\left\{\frac{a}{t}, \frac{a+t-1}{t}, \frac{m^2}{t}, \frac{m^2+t+1}{t}, 0\right\} = 0$.

## 7. Simulations and Numerical Results

In this part, the simulation results are obtained to analyze the effect of the detection method, atmospheric turbulence, pointing errors, relay mode, and fixed relay gain on the considered communication systems. Additionally, analytical expressions are validated by Monte Carlo simulations. The FSO link undergoes $\mathcal{F}$-distributed fading, in which the values of $a$ and $b$ can be calculated by (9) with the aperture diameter of the receiver $D = 10$ mm,

$l_0 = 5.98$ mm, $L_0 = 0.8$ m, $\lambda = 1550$ nm, and the distance $L$ is set to 3.5 km. We set $\gamma_{th} = 0$ dB. For strong and moderate turbulence, the refraction index structure parameter $C_n^2$ is $5 \times 10^{-14}$ m$^{-2/3}$ and $1.25 \times 10^{-14}$ m$^{-2/3}$, respectively, which results in $\sigma_1^2 = 4.0230$ and $\sigma_1^2 = 1.0058$. Hence, the atmospheric turbulence parameters are derived as follows: $a = 1.4753$, $b = 4.9821$ for strong turbulence and $a = 2.5190$, $b = 5.8804$ for moderate turbulence.

Figure 2 illustrates the outage probability under strong turbulence. The analytical expressions of the outage probability are based on (34) and (41). It is assumed that $\bar{\gamma}_{RF} = \bar{\gamma}_{FSO}$ and $C = 1$. The outage probability decreases as the value of $m$ changes from 1 to 6.7. For example, at 16 dB SNR in variable-gain relaying under heterodyne detection, the outage probability is $3.31 \times 10^{-2}$ with $m = 6.7$, while the outage probability increases to $6.59 \times 10^{-2}$ with $m = 1$. Moreover, it can be seen that the fixed-gain relaying offers superior performance compared with the variable-gain relaying under the IM/DD technique. For instance, at 28 dB SNR, the outage probability is $6.58 \times 10^{-2}$ for the variable-gain relaying, while the outage probability decreases to $6.65 \times 10^{-3}$ for the fixed-gain relaying. The reason is that the variable-gain relaying amplifies the noise and interference with the signal, while the relay gain is independent of the channel conditions for fixed-gain relaying. Additionally, the results also demonstrate that the system has better outage performance under HD as compared to IM/DD. For instance, at 28 dB SNR and under the situation of variable-gain relay with $m = 1$, the outage probability is $4.54 \times 10^{-3}$ under HD, while the outage probability increases to $6.58 \times 10^{-2}$ under IM/DD. Moreover, the asymptotic curves for all terms coincide with the analytical curves at high SNR values, and the simulation results match the analytical results.

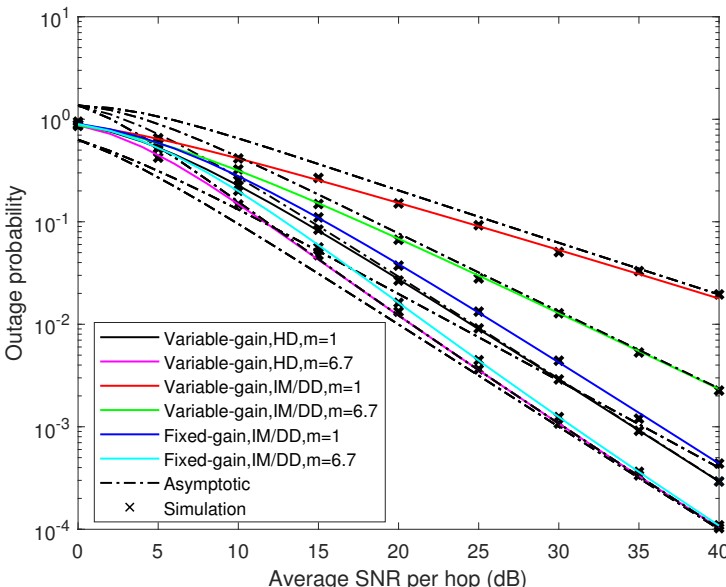

**Figure 2.** Outage probability performance under strong turbulence.

Figure 3 presents the outage probability performance with fixed-gain relaying under strong turbulence with $m = 1$ for various $C$. The outage probability performance becomes worse when the value of $C$ becomes larger (i.e., the fixed gain becomes smaller). For instance, the outage probability decreases from $4.46 \times 10^{-3}$ to $1.72 \times 10^{-3}$ when the value of $C$ decreases from 10 to 1 (fixed gain increases).

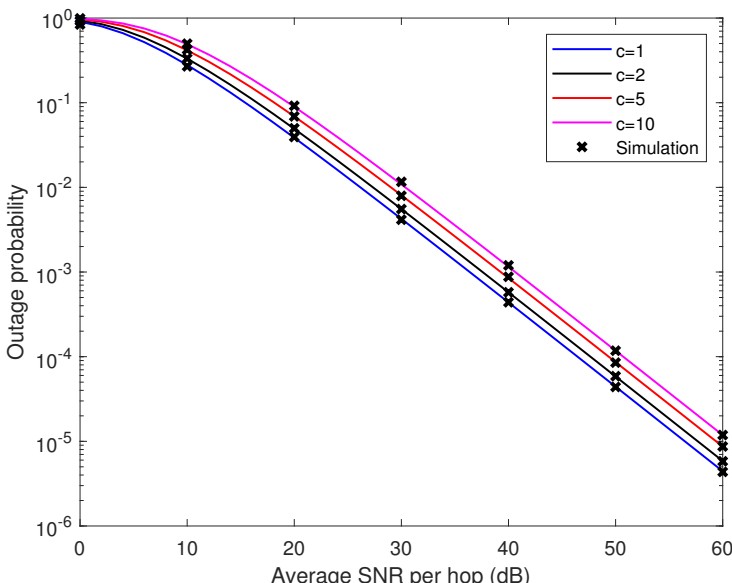

**Figure 3.** Outage probability performance under various *C*.

In Figure 4, the average BER performance under strong turbulence conditions with variable-gain relaying is presented. The analytical expressions for average BER are based on (37) and (43). The figure is plotted at $\bar{\gamma}_{FSO} = 20$ dB and $m = 1$. The figure shows that CBPSK offers the best performance compared with other binary modulation schemes, and NBFSK offers the worst performance as compared with the other three modulations. For instance, at 4 dB SNR, the average BER values for the four binary modulations are 0.2217 for NBFSK, 0.1422 for DBPSK, 0.1243 for CBFSK, and 0.0745 for CBPSK.

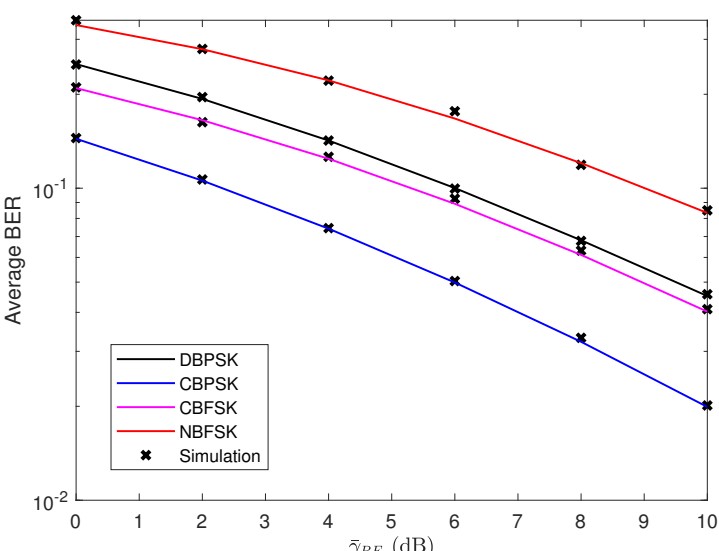

**Figure 4.** Average BER performance under variable-gain relaying and strong turbulence.

In Figure 5, the ergodic capacity performance under fixed-gain relaying is presented. The ergodic capacity expressions are based on (40) and (45). It is assumed that $\bar{\gamma}_{RF} = 10$ and $C = 1$. The figure shows that the HD method outperforms the IM/DD method. For example, at 6 dB SNR and $m = 6.7$, the ergodic capacity is 1.0219 bits/s/Hz for the IM/DD technique under strong turbulence, while the ergodic capacity significantly increases to 1.8193 bits/s/Hz for heterodyne detection under the same atmospheric turbulence. The ergodic capacity decreases as the pointing errors become severe. For example, the ergodic capacity decreases from 2.8938 bits/s/Hz to 1.6904 bits/s/Hz when the parameter *m*

decreases from 6.7 to 1 in the case of the HD method under strong turbulence at the 14 dB SNR. Moreover, the system performance becomes worse when the atmospheric turbulence becomes severe. For instance, under heterodyne detection, $m = 1$, and at 4 dB SNR, the ergodic capacity is 0.94833 bits/s/Hz in strong turbulence, while the ergodic capacity is 2.3058 bits/s/Hz in moderate turbulence. It shown that the simulation results closely match the analytical results.

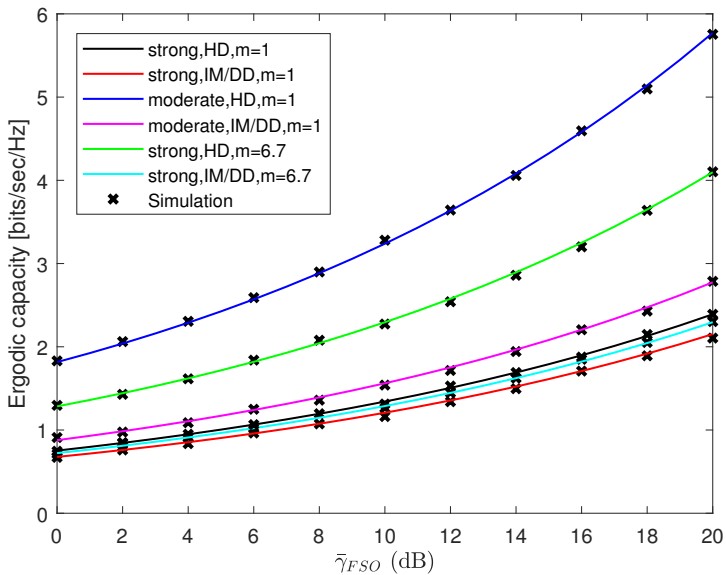

**Figure 5.** Ergodic capacity performance under fixed-gain relaying.

## 8. Conclusions

This paper provides performance analysis of a dual-hop mixed RF-FSO system with AF relaying, in which the RF link experiences Rayleigh fading and the FSO link follows $\mathcal{F}$-distributed fading with pointing errors. The CDF, MGF, and the moments of equivalent SNR are derived. Utilizing these formulas, the performance metrics are derived, such as outage probability, average BER, and ergodic capacity. Moreover, the asymptotic outage probability expressions for variable-gain relay and fixed-gain relay are obtained. The results indicate that the detection method, atmospheric turbulence, pointing errors, relay mode, and fixed relay gain impact the system performance. The system performance becomes worse when the pointing errors and atmospheric turbulence become severe. The system has better performance under HD compared to the IM/DD.

**Author Contributions:** L.H., X.L., Y.W. and B.L. contributed equally to this work. All authors have read and agreed to the published version of the manuscript.

**Funding:** This work was supported by Hebei Province Oversea Returnee Project under Grant No.C20190369.

**Institutional Review Board Statement:** Not applicable.

**Informed Consent Statement:** Not applicable.

**Data Availability Statement:** Not applicable.

**Conflicts of Interest:** The authors declare no conflict of interest.

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
