# Peer review of "Performance Analysis of Mixed Rayleigh and Distribution RF-FSO Cooperative Systems with AF Relaying"

_electronics, doi:10.3390/electronics11152299_

Round 1
Reviewer 1 Report
The idea presented and developed in the paper seems to be interesting and the results obtained are promising but I have the following concerns:
1) The abstract should be rewritten more precisely. Measurable results and conclusions should be added.
- 2) Lack of a reference to the Free Space Optics Communication. Recently, an interesting analysis concerning these issues was developed in the articles:
- Pang, X.; Ozolins, O.; Zhang, L.; Schatz, R.; Udalcovs, A.; Yu, X.; Jacobsen, G.; Popov, S.; Chen, J.; Lourdudoss, S, Free-Space Communications Enabled by Quantum Cascade Lasers. Phys. Status Solidi A 2021, 218: 2000407;
- Spitz, O.; Herdt, A.; Wu, J. et al. Private communication with quantum cascade laser photonic chaos. Nat. Commun. 2021, 12, 3327;
- Gaji´c, A.; Radovanovi´c, J.; Vukovi´c, N.; Milanovi´c, V.; Boiko, D.L. Theoretical approach to quantum cascade micro-laser broadband multimode emission in strong magnetic fields. Phys. Lett. A 2021, 387, 127007;
- Garlinska, M.; Pregowska, A.; Gutowska, I.; Osial, M.; Szczepanski, J. Experimental Study of the Free Space Optics Communication System Operating in the 8–12 μm Spectral Range. Electronic 2021, 10, 875;
- Lionis, A.; Peppas, K.; Nistazakis, H.E.; Tsigopoulos, A.D.; Cohn, K. Experimental Performance Analysis of an Optical Communication Channel over Maritime Environment. Electronics 2020, 9, 1109;
- Wang, Y.; Xu, H.; Li, D.; Wang, R.; Jin, C.; Yin, X.Y.; Gao, S.; Mu, Q.; Zuan, L.; Cao, Z. Performance analysis of an adaptive optics system for free-space optics communication through atmospheric turbulence. Sci. Rep. 2018, 8, 1124.
I recommend adding (including) into the paper a Paragraph, dealing with this issue among others on the above references.
1) The quantitative comparison with other existing literature algorithms (papers) should be added in Section: Discussion.
2) The measurable conclusion should be added.
3) The language should be carefully revised.
At this moment I would not recommend this paper for publication, I would recommend Major Revision.
Reviewer 2 Report
1. What are the paractical implicaton of this proposed system model?
2. Why the authors are considered FSO link for second hop why not RF?
3. The manuscript has written well but few places missing punctuation and indentation needed.
4. Section 4.1 and 4.1.1 are confusing, Please clarify it.
5. Reference number 31 should be more clear and it should be access online.
Round 2
Reviewer 1 Report
The authors took into account my comments and recommend this paper for publication.